# Study of Cytotoxic Properties of an Experimental Preparation with Features of a Dental Infiltrant

**DOI:** 10.3390/ma14092442

**Published:** 2021-05-08

**Authors:** Małgorzata Fischer, Anna Mertas, Zenon Paweł Czuba, Małgorzata Skucha-Nowak

**Affiliations:** 1Unit of Dental Propedeutics, Department of Conservative Dentistry with Endodontics, Faculty of Medical Sciences in Zabrze, Medical University of Silesia, 40-055 Katowice, Poland; mskuchanowak@sum.edu.pl; 2Department of Microbiology and Immunology, Faculty of Medical Sciences in Zabrze, Medical University of Silesia, 40-055 Katowice, Poland; amertas@sum.edu.pl (A.M.); zczuba@sum.edu.pl (Z.P.C.)

**Keywords:** dental materials, polymers, antimicrobial properties, infiltrants, metronidazole

## Abstract

Microinvasive dentistry is based on the treatment of early carious lesions with the use of dental infiltrants. The commercially available Icon dental infiltrant does not contain any bacteriostatic component. An experimental preparation enriched with the missing component was synthesised. The aim of this study was to evaluate the cytotoxicity of the experimental preparation. Mouse fibroblasts of the L-929 lineage were used for the in vitro study. Cell morphology and viability were assessed. In the cytotoxicity analysis, it was shown that the experimental preparation (42.8 ± 10.3) after 24 h at two-fold dilution showed similar cytotoxicity to Icon (42.7 ± 8.8) (*p* > 0.05), while at four-fold dilution experimental preparation (46.7 ± 3.1), it was less toxic than Icon (34.2 ± 3.1) (*p* < 0.05). The experimental preparation has the potential to provide an alternative to the Icon commercial preparation. Further research is needed to evaluate the cytotoxicity of the experimental preparation over a longer period of time.

## 1. Introduction

Microinvasive dentistry (also called microdentistry) is a modern preventive and therapeutic concept based on an attempt to preserve as much as possible of the patient’s healthy tissues and to postpone the onset of invasive (surgical) treatment, that is mechanical removal of carious lesions, in favour of therapeutic (biological) treatment—remineralisation of carious lesions along with modification of the oral cavity environment [1,2,3,4,5,6,7]. Currently, the development of technology is moving towards the application of stem cells in numerous fields of medicine. The repair and regeneration of tissue structures can provide an alternative to current treatment protocols. However, there are still no studies showing the reconstruction of enamel tissues and root cement with the participation of stem cells. Due to this fact, non-invasive treatment of early carious lesions on the surfaces of tooth tissues can be treated with the use of dental infiltrants [8]. This trend in conservative dentistry makes it possible to stop the progression of a disease at an early stage and to restore the normal structure and function of dental tissues.

A dental infiltrant is a substance based on low-viscosity polymer resins, capable of deep penetration of demineralised tooth tissues and of filling the inter-crystalline spaces of hard tissues. That process reduces porosity of the decalcified tissues. As a result, the carious spot becomes inaccessible for cariogenic bacteria and their toxins. The result of this process is a halt in the progression of caries’ development. Icon (DMG, Hamburg, Germany) is an example of a commercially available dental infiltrant preparation, which was launched in 2009. It is intended for treatment of carious lesions located on smooth and tangential tooth surfaces [9,10,11,12,13]. Icon is recommended for use in early-stage carious lesions to a maximum depth of D1 (Manji radiographic classification of lesion depth) [13,14]. In order to be referred to as a dental infiltrant, the preparation should feature low density, high surface tension, hydrophilicity, no toxic effects on the organism, bacteriostaticity, no interaction with food and drugs, ability to polymerise to a solid state, resistance to chemical and mechanical agents and have good aesthetics after polymerisation [9]. The advantages of the Icon commercial preparation include the ability to polymerise to a solid state, resistance to chemical and mechanical agents and its ability to mimic the tooth colour. Low viscosity and wetting angle, and high surface tension and hydrophilicity allow infiltrants to penetrate pores of decalcified enamel. However, the said preparation is not without its drawbacks. Icon does not contain a component responsible for inhibiting the multiplication of microorganisms on the tooth surface. The bacteriostatic nature of the drug is defined as the effect of the preparation on bacteria by inhibiting their proliferation and preventing the massive breakdown of bacterial cells with the release of antigens (including endotoxins) [15].

The fact that the commercial preparation Icon does not have an ingredient responsible for bacteriostaticity in its composition led the authors to undertake research towards the synthesis of an experimental preparation enriched with the missing component. The developed experimental preparation was enriched with PMMAn-MTZ monomer, which has adhesive properties. The action of metronidazole targets anaerobic and oxygen-deficient organisms. This chemotherapeutic agent is active against protozoa and, at the same time, displays antibacterial activity (against bacteria of the following genera: *Bacteroides*, *Fusobacterium*, *Eubacterium*, *Clostridium*, *Peptococcus*, *Peptostreptococcus*, among others). Metronidazole enters the cells as a prodrug, then it is converted into a short-term nitroso radical. The chemotherapeutic agent in this form is cytotoxic and acts by inhibiting DNA synthesis and damaging it through oxidation reactions. That process results in DNA strand breaks, which leads to degradation and death of the cell [16]. The synthesis of PMMAn with metronidazole provides a component that potentially inhibits microbial proliferation on the surface of dental hard tissue. PMMAn and metronidazole are linked by an ester bond, which is hydrolysed when exposed to water (saliva). In this way, metronidazole can inhibit bacterial growth on the tooth surface in a continuous manner.

One of the basic tests of new materials and therapeutic preparations which are to be used in medicine in the future is to determine the cytotoxic activity of a given substance. Determination of cytotoxicity is the basis for the assessment of the action of the drug and whether its influence on the human body is safe or not. Literature studies show that the development of molecular research enables a better understanding of the aetiology of diseases, complementing clinical practice. The use of profile gene expression can facilitate and replace classical techniques of cytotoxicity analysis. This method can provide much more information than classical methods [17]. Among the available methods for assessing the cytotoxicity of the tested preparations, there are tests that allow direct or indirect measurement of various parameters related to the physiology of cells, against which the activity of a given material is tested, including integrity of the cell membrane, proliferation, activity of cellular metabolic enzymes, as well as the amount of protein or genetic material [18]. There are numerous tests, for example, the test to assess the activity of the lactate dehydrogenase (LDH—lactate dehydrogenase test), the test based on the uptake of neutral red by lysosomes (NR—neutral red test), the test to demonstrate the presence of the lysosomal enzyme N-acetyl-beta-D-glucosaminidase in culture centrum (NAG test), the test determining the number of cells using sulforhodamine (SRB—sulforhodamine test) and one of the most commonly used tests—the MTT test (3-(4,5-dimethylthiazol-2-yl)-2,5-diphenyltetrazolium bromide test)—using the activity of cellular metabolic enzymes [19]. This method allows to estimate the percentage of living cells in cultures contacted with the tested preparation or its extract by measuring mitochondrial dehydrogenase activity, thus making it possible to determine the cytotoxicity of the tested preparation. Based on the recommendations of EN ISO 10993-5 standard: “Biological evaluation of medical devices—Part 5: In vitro cytotoxicity testing”, a reduction in the viability of cells contacted with test preparation extracts greater than 30%, when compared to a control cell culture, is considered to have a cytotoxic effect. To obtain a complete cytotoxicity profile for a given substance, tests are performed with different concentrations/dilutions of test extracts [20].

A small sample of literature reports on the subject of cytotoxicity in the area of infiltrants prompted the authors to undertake research in this field [21].

The main aim of our study is to assess the morphology and viability of cells of the model line of mouse fibroblasts (L-929) contacted with extracts of preparations with the characteristics of a dental infiltrant (experimental and commercial preparation).

The first null hypothesis is that the developed experimental preparation with the addition of metronidazole has all the properties of a dental infiltrant. The second null hypothesis is that the developed formulation after 24 h at 2-fold dilution and 4-fold dilution shows at least the same cytotoxicity as the commercial preparation of Icon.

## 2. Results

Comparing the mean viability of L-929 cells (%) after 24 h of incubation with a 2-fold dilution of the extracts of the tested preparations, it was found that there was no statistically significant difference (*p* > 0.05) between the experimental preparation (42.8 ± 10.3) and the Icon commercial preparation (42.7 ± 7.8), as shown in Table 1.

Comparing the mean viability of L-929 cells (%) after 24 h of incubation with a 4-fold dilution of the extracts of the tested preparations, it was found that the obtained values were statistically significantly different (*p* < 0.05) between the experimental preparation (46.7 ± 3.1) and the Icon commercial preparation (34.2 ± 3.1), as shown in Table 1 and graphically depicted in Figure 1.

In the next part of the statistical analysis, the mean cell viability of line L-929 (%) after 24 h of incubation with the tested extracts was compared according to the dilution multiplicity of the extracts of the tested preparations. The mean cell viability (%) for each preparation is statistically significantly different in relation to the dilution multiplicity. The experimental preparation in 2-fold dilution (42.8 ± 10.3) and in 4-fold dilution (46.7 ± 3.1) are statistically different (*p* < 0.05). The Icon preparation in 2-fold dilution (42.7 ± 7.8) and in 4-fold dilution (34.2 ± 3.1) are statistically different (*p* < 0.05). That data is shown in Table 1.

The obtained extracts of the tested preparations and the evaluation of L-929 cell viability by MTT assay are shown in Figure 2. Visible intensity of staining after the addition of DMSO is directly proportional to the cell viability.

During microscopic observations, images of L-9329 cells obtained for the experimental preparation were evaluated according to EN ISO 10993-5 standard and compared with control cultures of adhered L-929 cells, as shown in Figure 3.

## 3. Discussion

Microinvasive dentistry presents an innovative model for the treatment of early carious lesions. It is used in numerous branches of dentistry, including in the field of conservative dentistry, dental prosthetics and orthodontics. In the field of prosthetics, thanks to the use of milled and moulded mock-ups, tissue preparation for permanent restorations enables the preservation of more healthy tooth tissues, and in the area of conservative dentistry and orthodontics, e.g., after orthodontic treatment, in areas located in the vicinity of orthodontic brackets, where plaque has been deposited for a long time and tissue demineralisation occurs [22,23,24]. Mattousch et al. showed in an in vivo study that white spots formed after orthodontic treatment have a limited capacity for spontaneous remineralisation after removal of braces [25]. Infiltration of carious lesions effectively reduces the progression of lesions not only on smooth tooth surfaces but also in the interdental area. Aesthetics is an important aspect of modern dentistry, and infiltration allows for minimally interventional treatment of lesions while maintaining the continuity of the patient’s own tissues, which definitely has a positive effect on the final result in terms of aesthetics of the treated area. Studies available in the literature have shown that not only Icon infiltrant, but also the composite sealers Opiguard (Kerr, Orange, CA, USA) and PermaSeal (Ultradent, South Jordan, UT, USA) applied on the surface of early carious lesions proved to be able to infiltrate and, moreover, showed similar improvements in terms of aesthetics, and the colour stability of the infiltrated white spots lasted for at least 2 months [26,27]. That issue was also addressed by Paris et al. and Yuan et al., who found that colour change and masking of early carious lesions can be achieved with use of infiltrants whose RI (refractive index) is close to that of enamel [28,29].

Schmidlin et al. proved in an in vitro study that an infiltrant alone does not prevent enamel demineralisation because it does not have an antimicrobial component [30]. Additionally, Skucha-Nowak et al. observed that the Icon commercial preparation does not have a component that would confer bacteriostatic properties to dental infiltrants, affecting the inhibition of microbial proliferation on the tooth surface [31]. This fact prompted the authors to undertake research in this field.

Metronidazole is a highly active agent with a wide range of action against Gram-negative anaerobic bacteria, as well as Gram-positive anaerobic bacteria [32]. This agent is used, among others, in periodontal practice to reduce anaerobic microflora in pockets of over 5 mm depth. Topical use of metronidazole is beneficial due to its ability to release antimicrobials in concentrations high enough to affect pathogens, even in subgingival biofilms, yet far less than those administered systemically, which prevents a build-up of resistance [33]. Mombelli et al. observed a beneficial clinical and microbiological effect after topical application combined with ornidazole and thymidazole [34]. Other researchers report that topical metronidazole is not fully safe, as there are no studies on adverse effects during long-term metronidazole use [35]. On the other hand, in the study by Kida et al., the release of metronidazole from hydrogel dental dressings follows first-order kinetics, where the reaction rate is proportional to the substrate concentration and its value decreases with time [36]. In their study, Skucha-Nowak et al. demonstrated that the inclusion of a metronidazole component in the composition of an experimental preparation does not affect the process of polymerisation of a preparation by light. They proved that work with a preparation enriched with a bacteriostatic component does not differ in quality from work with a commercially available preparation, Icon [9]. A similar study was performed by Collares et al., who synthesised a preparation with the characteristics of a dental infiltrant by adding polyhexamethylene guanidine hydrochloride (PHMGH) in the amounts of 0.5 and 1 wt%, which is responsible for antibacterial activity against *Streptococcus mutans* and progression of carious lesions. Its mechanism of action is related to electrostatic interaction. The addition of PHMGH reduced the growth and adhesion of carious bacteria on the surface of resin at both tested concentrations, without affecting the physical properties of this material, i.e., conversion rate, wetting angle or surface free energy. However, the researchers did not evaluate the cytotoxic effect of the synthesised formulation, therefore making it impossible to assess its use in vivo and to compare the effect of the applied antimicrobial component on the cells in relation to our antimicrobial component, metronidazole [21]. Zhang et al. synthesised a dental adhesive with an antibacterial component containing dimethylaminododecyl methacrylate (DMADDM), affecting *Streptococcus mutans*, *Streptococcus gordoni* and *Streptococcus sanguis* [37]. In the present study, the authors used an antibacterial component containing metronidazole, which shows broad-spectrum antibacterial activity against *Bacteroides*, *Fusobacterium*, *Eubacterium*, *Clostridium*, *Peptococcus*, *Peptostreptococcus* and protozoa. The supragingival plaque is mainly formed by Gram-positive streptococci, including *Streptococcus mutans*, *Streptococcus salivarius*, *Streptococcus mitis* or *Lactobacillus*, while the subgingival plaque is mainly dominated by Gram-negative anaerobic bacteria, such as *Fusobacterium nucleatum* and *Porphyromonas gingivalis* [38]. The groups of Fu et al. and Wu et al. also studied the influence of the antimicrobial component in the composition of composite resins. According to the researchers, the use of nano-MgO shows a broad antibacterial spectrum against fungi (*Candida albicans*), viruses and bacteria (*Staphylococcus aureus*, *Enterococcus faecalis*, *Escherichia coli*) [39]. The use of an antimicrobial component in the composition of an experimental preparation used for infiltration in our study may potentially inhibit the development of caries in the place where it was applied.

The rapid development of technology means that more and more new preparations for use in dentistry are created.

Mechanical properties are among the key factors when selecting a dental material for use by a dentist.

Chieruzzi et al. undertook an assessment of the mechanical properties of glass-ionomer materials. They also enriched their study with the assessment of the antibacterial effect of a glass-ionomer preparation enriched with the following ingredients: nanohydroxyapatite (Sealent, Miromed Srl, Milano, Italy), ciprofloxacin antibiotic (Ciproxin, Bayer SpA, Milano, Italy) and MDA—zinc L-carnosine (Hepilor, Azienda Farmaceutica Italiana, Ascoli Piceno, Italy) [40]. Khosravani also undertook an evaluation of the mechanical properties of composite materials. In his research, he showed that a greater number of cracks and roughness may result in increased adhesion of bacteria [41].

Modern dentistry mainly uses light-curing composite materials to fill hard tissue defects. These materials include monomers, initiators, retarders and photo-stabilisers, which may potentially cause cytotoxic effects. Most dental composites are based on bis-GMA and TEGDMA. The TEGDMA monomer is used to increase the conversion of vinyl groups during polymerisation. However, an increase in polymerisation shrinkage occurs during the process, causing stress in the composite, which weakens the seal between it and the tooth structure [42]. While evaluating the properties of experimental formulations, other researchers have shown that the addition of hydrophobic monomers and solvents to TEGDMA mixtures does not improve the depth of penetration of the caries-affected enamel surface. However, it affects the degree of conversion and the modulus of elasticity [43]. Inamitsu et al. observed that HEMA monomer and TEGDMA inhibit osteoclast differentiation [44]. Additionally, other researchers have shown that TEGDMA decreases the differentiation capacity of odontoclasts in pulp cells and the differentiation rate of odontoclasts increases proportionally to the amount of TEGDMA monomer [45]. Yoshii in his study demonstrated that bis-GMA is the most cytotoxic compound, followed by UDMA, TEGDMA and finally MMP [46]. The literature describes the possibility of reducing the cytotoxic effect of complex materials by diluting the substance during its penetration through biological in vitro barriers [43]. Moreover, the present research proved that a 4-fold dilution used immediately after the application of the experimental preparation reduces the cytotoxic effect. The results available in the literature have shown that resin-based materials can adversely affect oral tissues, because the release of residual resin monomers can occur up to several hours after application, as a result of incomplete polymerisation. Monomers released from resin-based materials can enter the surrounding tissues, especially when these materials are used in close proximity to gingival mucosa, exerting adverse effects and inducing apoptosis of various types of periodontal cells [47]. The cytotoxicity of infiltrants was studied by Golz et al., who noted that although Icon resin infiltration is a microinvasive technique, it can affect pulp inflammation, as HEMA and TEGDMA, due to their lipophilic nature, have the ability to penetrate the lipid layer of the cell, and in vivo models have demonstrated their presence in the dental pulp. Golz et al. also observed that reduction of the curing time of resin-based composites increases the index of cell apoptosis. In the present study, the experimental preparation with an antimicrobial component in the form of metronidazole also contains HEMA, TEGDMA and CQ in its composition [48]. Our results may suggest that in order to reduce the cytotoxic effect of the experimental preparation on the surrounding soft tissues, it is important to rinse the oral cavity thoroughly with water after its application. 

Many practicing dentists pay special attention to the way the preparation is used. This has the effect of interacting with the surrounding tissues in the oral cavity. The contact of the preparation with oral fluids and the lack of the need to isolate the application site makes the procedure much easier for the dentist.

On the basis of their own research, the authors concluded that the developed experimental preparation with metronidazole can be potentially used as a dental infiltrant in dentistry.

The authors also proved that the experimental preparation is characterised by a similar cytotoxicity after 24 h and 2-fold dilution as the commercial preparation Icon, while at 4-fold dilution, it is statistically significantly lower.

The influence of the daily activities performed by the patient in the oral cavity is also important. A question may be asked as to how daily hygienic procedures, such as tooth brushing, the influence of food consumption as the years go on and time may influence the degree of cytotoxicity of the experimental preparation.

Further research should be conducted.

## 4. Materials and Methods

### 4.1. Experimental Preparation

An experimental preparation with the characteristics of a dental infiltrant, enriched with a component responsible for bacteriostaticity (metronidazoles), was used in the study, as shown in Table 2 [11]. The experimental preparation formed the research group. A freely available Icon commercial preparation was used in the control group [7].

### 4.2. Polymerisation of Preparations

A wireless LED polymerisation lamp C01-C Premium Plus (Premium Plus International Limited, Bournemouth, UK) was used for the polymerisation process. It emits light with a wavelength between 440 and 480 nm. The length of the optical fibre cable is 8 mm. Full mode with a radiation power of 1200 mW/cm^2^ was used for the study (Figure 4).

### 4.3. Obtaining Extracts of the Tested Preparations

A 10 µL volume of the test preparation was dispensed to the bottom of a 96-well microtiter plate. The test preparations were polymerised, and the microtiter plate was sterilised. Subsequently, 200 µL of RPMI 1640 medium with 10% FBS was dispensed into the wells with the polymerised preparations to obtain extracts of the tested preparations. The culture medium was also placed in the wells without polymerised preparations in order to obtain the medium for further studies, after the 24 h incubation, in the conditions corresponding to those in which extracts of the tested preparations were made. After 24 h of incubation at 37 °C in an atmosphere containing 5% of CO_2_ at 100% relative humidity, both extracts and the culture medium used in the next stage of the study to assess cell viability L-929 were ready (Figure 4).

### 4.4. Evaluation of Cytotoxicity of the Tested Preparation Extracts

Cytotoxicity assessment of the test formulations was performed according to the recommendations of the EN ISO 10993-5 standard, with the use of mouse fibroblast line L-929 (NCTC clone 929) purchased from the American Type Culture Collection (Manassas, VA, USA). The L-929 cell line (ATCC, catalogue number CCL-1) consisted of mouse subcutaneous connective tissue fibroblasts of the C3H/An strain. Under in vitro culture conditions, they were directly contacted with extracts of the test preparations and cell morphology was successively assessed by microscopy, while cell viability was tested with the MTT assay. RPMI 1640 medium with 10% addition of heat-inactivated foetal bovine serum (FBS), penicillin (100 IU/mL) and streptomycin (100 µg/mL) were used to culture L-929 cells (mouse subcutaneous connective tissue fibroblasts). The culture was performed in 25 cm^2^ polystyrene adherent cell culture bottles (Nunc EasYFlasksTM NunclonTMDelta by Nunc A/S, Roskilde, Denmark). The cells were continuously cultured in an MCO-17 AIC incubator by Sanyo (Osaka, Japan), which provided stable culture conditions (temperature 37 °C, atmosphere containing 5% CO_2_ at 100% relative humidity). The cells were passaged at intervals of 2–3 days. A suspension with a final density of 1 × 105 cells/mL of medium was used for experimental studies. The density of the cell suspension was assessed by microscopy using a Bürker chamber. The cells of the model line L-929 (mouse fibroblasts) under in vitro culture conditions were contacted with previously obtained 24 h extracts of the tested preparations diluted with a 2-fold or 4-fold ratio in the culture medium. The test was conducted with the Norm PN-EN ISO 10993-5, in conditions corresponding to the conditions of the oral cavity. After 24 h of incubation, cell morphology (microscopic method) and cell viability (MTT assay) were evaluated [20].

The stages of the laboratory procedure are presented graphically in Figure 4.

#### 4.4.1. Microscopic Observations

An Olympus IX51 inverted fluorescence microscope with a video track with an image recording camera (OLYMPUS, Tokyo, Japan) was used for microscopic observations. During microscopic observation of cells, a magnification of 200 times was used.

#### 4.4.2. Viability Assessment of L-929 Cells Contacted with Extracts of the Tested Preparations

The 3-(4,5-dimethylthiazol-2-yl)-2,5-diphenyltetrazolium bromide test (MTT assay test), conducted by measuring mitochondrial dehydrogenase activity, allowed to assess the percentage of viable cells in cultures contacted with the test extract, thus determining the cytotoxicity of the test preparation. A 200 µL volume of each suspension of the L-929 line cells at a density of 1 × 105 cells/mL in the RPMI 1640 medium with 10% FBS (i.e., 20,000 cells/well) was dispensed into the wells of a 96-well microplate. After 24 h of incubation at 37 °C in the atmosphere containing 5% CO_2_ at 100% relative humidity, the supernatant was removed, and 200 µL of the 24 h extract of the specific test preparation, diluted two or four times in the culture medium or the medium with adhered cells after 24 h of incubation, conducted in parallel with obtaining the extracts, were added to the wells, respectively. Control cultures consisted of cells contacted with fresh culture medium.

After 24 h of incubation at 37 °C in an atmosphere containing 5% CO_2_ at 100% relative humidity, the viability of L-929 cells was assessed by the MTT assay. For this purpose, a MTT solution with a final concentration of 1.1 mM in the culture medium was dispensed into each well after the culture medium had been removed. After 3 h of incubation at 37 °C in a CO_2_-free atmosphere and at constant relative humidity, the filtrate was removed, and 200 µL of each DMSO was added to the test and control cultures to extract MTT formazan. After 20 min, 150 µL of solution was collected from each well and its absorbance was determined at 550 nm using an Eon automated plate reader (BioTek Instruments, Winooski, VT, USA). The intensity of the purple colour of the solution was directly proportional to the amount of formazan formed and thus the number of living cells.

Cell viability, expressed in %, was calculated using the following formula:Cell viability (%) = A_b_/A_k_ × 100%
A_b_—absorbance of the test sample,A_k_—absorbance of the control sample.

### 4.5. Statistical Analysis

The viability of L-929 cells after 24 h of in vitro incubation with 2-fold (or 4-fold) diluted extracts of the experimental preparation and the Icon preparation were examined with the use of statistical analysis methods. At the beginning of the statistical analysis, basic measures of descriptive statistics, i.e., mean, median, minimum, maximum, standard deviation, coefficient of variation, skewness and kurtosis, were calculated. The part of the statistical analysis concerning the significance of the differences began with checking the normality of feature distributions in the tested samples using the Shapiro–Wilk test. Subsequently, the following statistical tests were applied: test of equality of variances (Levene’s test), test of equality of feature distributions for two independent samples (Mann–Whitney U test, Kruskal–Wallis H test), test of equality of feature means for two dependent samples (*t*-test for two dependent samples) and test of equality of feature means for two independent samples (*t*-test for two independent samples). All hypotheses were verified at the significance level of 0.05. Statistical software was used in the statistical analyses.

## 5. Conclusions

The experimental preparation has the potential to provide an alternative to the Icon commercial preparation.

Further research should be conducted to evaluate the cytotoxicity of the experimental preparation over a longer period of time.

## Figures and Tables

**Figure 1 materials-14-02442-f001:**
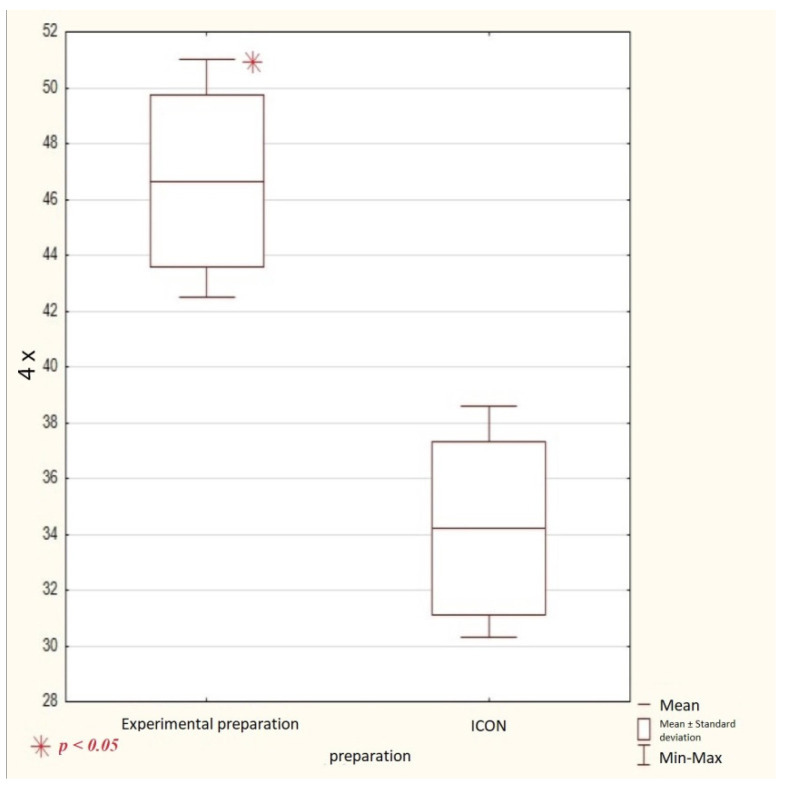
Comparison of mean viability of L-929 cells (%) after 24 h of incubation with a 4-fold dilution of the extract of the experimental preparation and the Icon preparation (*p* < 0.05). * statistical significance.

**Figure 2 materials-14-02442-f002:**
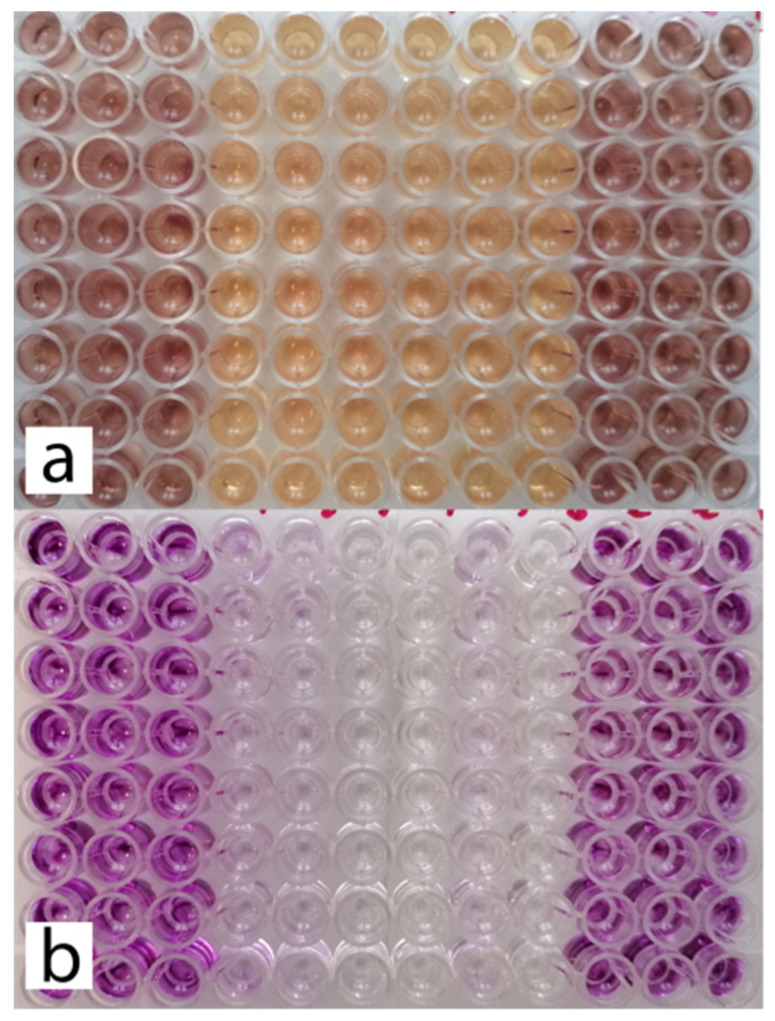
(**a**) Plate with cultures of L-929 cells immediately after a 24 h incubation period with 2-fold diluted extracts of the test preparations. (**b**) MTT assay performed for cultures shown in image (**a**) (after formazan extraction with DMSO).

**Figure 3 materials-14-02442-f003:**
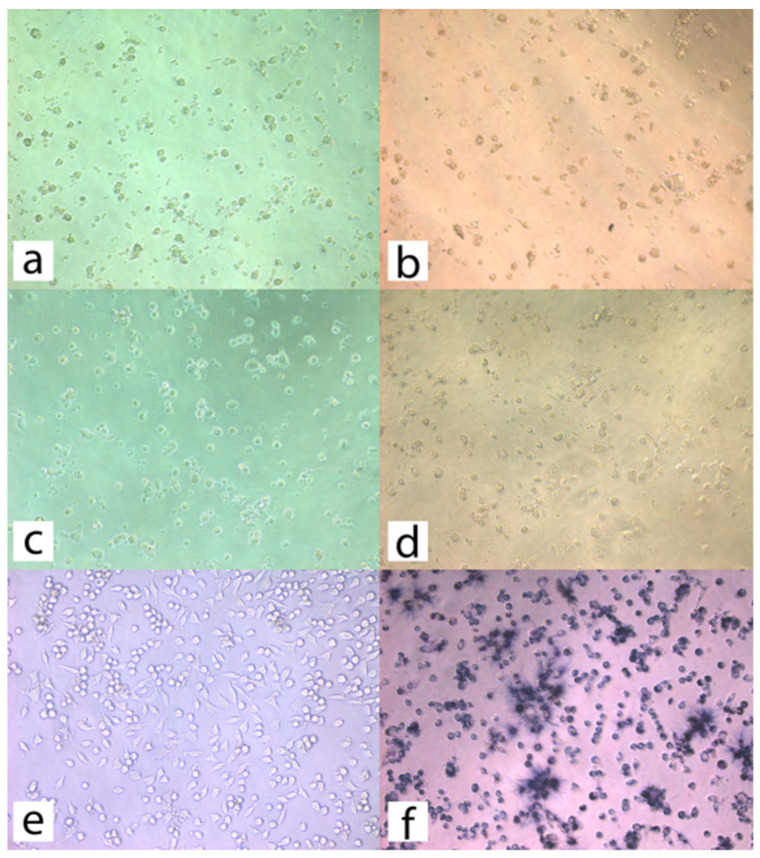
(**a**) L-929 line cells after 24 h incubation with a 2-fold diluted 24 h extract of the experimental preparation, (**b**) L-929 line cells after 24 h incubation with a 2-fold diluted 24 h extract of the experimental preparation after MTT assay, (**c**) L-929 cells after 24 h incubation with 2-fold diluted 24 h Icon extract, (**d**) L-929 cells after 24 h incubation with 2-fold diluted 24 h Icon extract after MTT test. (**e**) Control culture of adhered L-929 cells, (**f**) control culture of adhered L-929 cells after MTT test.

**Figure 4 materials-14-02442-f004:**
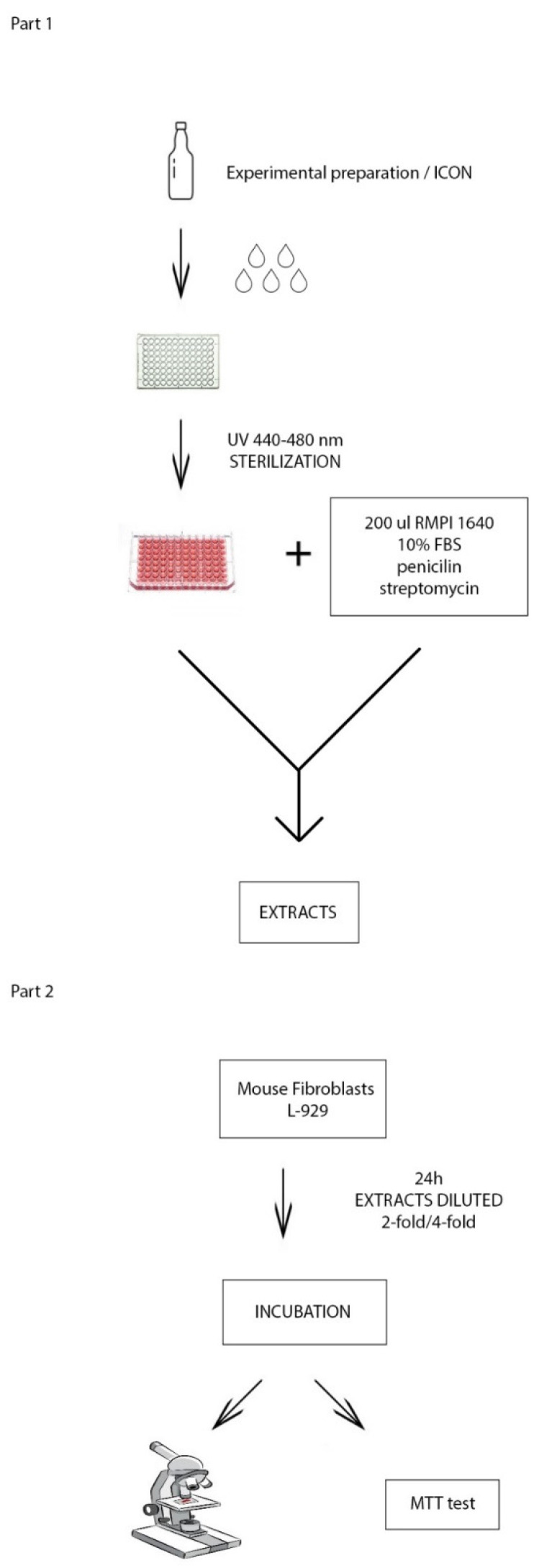
(**Part 1**) Scheme for obtaining 24 h extracts. (**Part 2**) Scheme for assessing cell viability.

**Table 1 materials-14-02442-t001:** Comparison of the mean viability of L-929 cells (%) after 24 h of incubation with a 2-fold and 4-fold dilution of the extract of the experimental preparation and the Icon preparation.

	Mean Viability of Cell (%)		
Preparation	2-Fold Dilution	4-Fold Dilution	
Experimental preparation	42.8 ± 10.3	46.7 ± 3.1	* *p* < 0.05
Icon	42.7 ± 7.8	34.2 ± 3.1	* *p* < 0.05
	* *p* > 0.05	* *p* < 0.05	

* statistical significance.

**Table 2 materials-14-02442-t002:** Composition and percentage content of the experimental preparation.

Component	Quantity (g)	Content (%)
TEGDMA	3.75	75
HEMA	1.25	25
PMMAn-MTZ *	0.05	1 *
DMAEMA *	0.05	1 *
CQ *	0.025	0.5 *

* Ratio to total mass of monomers. TEGDMA (triethylene glycol dimethacrylate, Fluka, Buchs, Switzerland); HEMA (2-hydroxyethyl methacrylate, Acros, Belgium, NJ, USA); PMMAn (2-(7-methyl-1,6-dioxo-2,5-dioxa-7-octenyl) trimellitic anhydride); MTZ (metronidazole, Acros, New Jersey, NJ, USA); DMAEMA * (N,N-dimethylaminoethyl methacrylate, Merck, Darmstadt, Germany); CQ * (camphorquinone, Aldrich, St. Louis, MO, USA).

## Data Availability

Department of Conservative Dentistry with Endodontics, Unit of Dental Propedeutics, Faculty of Medical Sciences in Zabrze, Medical University of Silesia in Katowice, Plac Akademicki 17, 41-902 Bytom, Poland; Tel.: +48-322-827-942.

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
