# Peer review of "Study of Cytotoxic Properties of an Experimental Preparation with Features of a Dental Infiltrant"

_materials, 2021, doi:10.3390/ma14092442_

Round 1

Reviewer 1 Report

There are some weaknesses through the manuscript which need improvement. Therefore, the submitted manuscript cannot be accepted for publication in this form, but it has a chance of acceptance after a major revision. My comments and suggestions are as follows:

1- Abstract gives information on the main feature of the performed study, but some details about the obtained results must be added.

2- Authors must clarify necessity of the performed research. Main objectives of the study must be clearly mentioned in introduction.

3- The literature study must be enriched. In this respect, authors must read and refer to the following papers: (a) https://doi.org/10.1016/j.jmbbm.2019.02.009 (b) https://doi.org/10.1016/j.dental.2019.08.044

4- It would be nice, if authors prepare the table with care (column width and color).

5- Since this manuscript deals with experimental study, authors must add some figures (real or schematic) to show concept and some conditions.

6- All figures and curves must be illustrated in a high quality (texts in some curves are illegible).

7- Standard deviation is the presented results must be discussed.

8- In its language layer, the manuscript should be considered for English language editing. There are sentences which have to be rewritten.

9- The conclusion must be more than just a summary of the manuscript. List of references must be updated based on the proposed papers. Please provide all changes by red color in the revised version.

Author Response

Dear Reviewer 1,

I would like you to thank you for your valuable comments and criticism regarding our article.

Thank You for giving us an very interesting proposal for a research paper that enriched our manuscript. 

As suggested by the Reviewer 1 and Reviewer 2, the manuscript has been modified.

Below is a Word file with answers. 

Reviewer 2 Report

This is an interesting work about filler cytotoxicity using MTT method. Some criticism are present:

The abstract section must be completely rewritten. All the first sentences are too discursive and long.

-Lines 26-29 absolutely useless to report all this information in this section. Reduce

-At the end of the abstract section, a sentence on the possible clinical effects of the trial must be added

-In the introduction section, when mentioning the MTT method, it is necessary to list, even if only for citation, the other methods of investigation alternative to this one.

-How come only 24 hours of observation were considered? Wouldn't it have been the case to observe longer-term effects?

-Moreover, it is not clear why only dry conditions were considered and not in humid environments, as it actually happens in the oral cavity.

-Tables from 1 to 3 must be merged into one only, and do the same with the relative figures. Too much uniform only brings confusion to the reader of the study

-The discussion section lacks a more complete narrative on the various forms of cytotoxicity found in various restorative materials in dentistry. In this regard, I recommend that you insert the following scientific work in the reference section which could be of help to the reader:

Chieruzzi, M .; Pagano, S .; Lombardo, G .; Marinucci, L .; Kenny, J.M .; Torre, L .; Cianetti, S. Effect of nanohydroxyapatite, antibiotic, and mucosal defensive agent on the mechanical and thermal properties of glass ionomer cements for special needs patients. J. Mater. Res. 2018, 33, 638–649.

-A section of the study limits and future perspectives is missing

-An extensive and certified correction of English grammar is absolutely required by this reviewer

Author Response

Dear Reviewer 2,

I would like you to thanky you for your valuable comments and criticism regarding our article.

Thank You for giving us an very interesting proposal for a research paper that enriched our manuscript.

As suggested by Reviewer, the manuscript has been modified. 

Below is a Word file with anwers.

Round 2

Reviewer 1 Report

The paper has been improved and corresponding modifications have been conducted. In my opinion, the current version can be considered for publication.

Author Response

I would like to thank you for all your valuable comments.

Reviewer 2 Report

All comment Were added 

Author Response

Thank you very much for all your valuable comments.
